# Removal of developmentally regulated microexons has a minimal impact on larval zebrafish brain morphology and function

Caleb CS Calhoun, Mary ES Capps, Kristie Muya, William C Gannaway, Verdion Martina, Claire L Conklin, Morgan C Klein, Jhodi M Webster, Emma G Torija-Olson, Summer B Thyme*[†]

Department of Neurobiology, The University of Alabama at Birmingham Heersink School of Medicine, Birmingham, United States

## eLife Assessment

This **important** work examines how microexons contribute to brain activity, structure, and behavior. The authors find that loss of microexon sequences generally has subtle impacts on these metrics in larval zebrafish, with few exceptions. The evidence is **solid**, using modern high-throughput phenotyping methodology in zebrafish. Overall, this work will be of interest to neuroscientists and generate further studies of interest to the field.

**\*For correspondence:**
summer.thyme@umassmed.edu

**Present address:** [†]Department of Biochemistry and Molecular Biotechnology, UMass Chan Medical School, Worcester, United States

**Abstract** Microexon splicing is a vertebrate-conserved process through which small, often in-frame, exons are differentially included during brain development and across neuron types. Although the protein sequences encoded by these exons are highly conserved and can mediate interactions, the neurobiological functions of only a small number have been characterized. To establish a more generalized understanding of their roles in brain development, we used CRISPR/Cas9 to remove 45 microexons in zebrafish and assessed larval brain activity, morphology, and behavior. Most mutants had minimal or no phenotypes at this developmental stage. Among previously studied microexons, we uncovered baseline and stimulus-driven phenotypes for two microexons (meA and meB) in *ptprd* and reduced activity in the telencephalon in the *tenm3* $B_0$ isoform. Although mild neural phenotypes were discovered for several microexons that have not been previously characterized, including in *ppp6r3*, *sptan1*, *dop1a*, *rapgef2*, *dctn4*, *vti1a*, and *meaf6*. This study establishes a general approach for investigating conserved alternative splicing events and prioritizes microexons for downstream analysis.

## Introduction

Alternative splicing greatly expands proteome diversity and can impact brain development and disease (*Vuong et al., 2016*). As most metazoan proteins are represented by multiple isoforms, defining the functional significance of individual splicing events is a vast challenge. Mouse studies do not have the throughput to assess the neural role of 10 or 100 s of isoforms, and cell culture does not recapitulate the complex development of an animal brain.

One class of particularly conserved and developmentally regulated spliced exons is microexons or mini-exons. These small exons, defined here as 3–30 nucleotides or 1–10 amino acids in length, are either gained or lost in transcripts in the brain compared to other tissues. While most are in-frame, this pathway can also trigger nonsense-mediated decay to influence protein levels (*Lin et al., 2020*). Although these neuron-specific exons were recognized almost 40 years ago (*Brugge et al.,*

1985; *Martinez et al., 1987*), their discovery and annotation have continued into recent years (*Parada et al., 2021*). Thus far, hundreds have been identified. Although more than one splicing regulator has been implicated in the alternative inclusion of microexons in neurons (*Ciampi et al., 2022*; *Li et al., 2015a*), many are controlled by the vertebrate-specific protein SRRM4 (*Irimia et al., 2014*; *Quesnel-Vallières et al., 2015*). Srrm4 interacts with Srsf11 and Rnps1 at polypyrimidine sequences upstream of microexons to regulate their inclusion (*Gonatopoulos-Pournatzis et al., 2020*).

Proteins that regulate and contain microexons are involved in neurodevelopmental disorders. In the brains of individuals with autism spectrum disorder (ASD), the level of SRRM4 is reduced, leading to the misregulated splicing of multiple microexons (*Irimia et al., 2014*). Mice haploinsufficient for Srrm4 display reduced social interaction, sensory hypersensitivity, and impaired synaptic transmission (*Quesnel-Vallières et al., 2015*). Furthermore, Srrm4 is involved in regulating the ASD-relevant KCC2-dependent GABAergic excitatory to inhibitory switch (*Nakano et al., 2019*), and a mouse model of ASD with disrupted *Pten* has decreased Srrm4 expression and microexon inclusion (*Thacker et al., 2020*). Loss-of-function variants in its interacting partner SRSF11 have been implicated in ASD in humans (*Yuen et al., 2017*). While the behavioral roles of only a few microexons have been studied in detail (*Gonatopoulos-Pournatzis and Blencowe, 2020*), the elimination of one from Eif4g1 altered synaptic plasticity and social behavior in mice (*Gonatopoulos-Pournatzis et al., 2020*). Many genes containing microexons also lead to neurodevelopmental disorders when mutated in humans. Examples include *CASK*-mediated microcephaly and cerebellar hypoplasia, *SPTAN1*-mediated childhood-onset epilepsy, and *MAPK8IP3*-mediated intellectual disability (*Giacomini et al., 2021*; *Platzer et al., 2019*; *Syrbe et al., 2017*). Taken together, this splicing program, the genes with microexons, and microexons themselves are important to enable proper neural development.

Although most microexons are unstudied, detailed structural and functional characterization of a small subset has revealed involvement in synapse development. The most well studied are found in the receptor-type tyrosine phosphatase (e.g., *ptprf*) and teneurin (e.g., *tenm4*) gene families. Recognized in the 1990s (*O'Grady et al., 1994*; *Pulido et al., 1995*), two microexons in presynaptic PTPδ (*Ptprd*), PTPσ (*Ptprs*), and LAR (*Ptprf*) modulate *trans*-synaptic interfaces and determine selective interaction with postsynaptic partners (*Lin et al., 2018*; *Li et al., 2015b*; *Park et al., 2020*; *Um et al., 2014*; *Yamagata et al., 2015*; *Yim et al., 2013*). Similarly, differential inclusion of two conserved microexons in teneurins can modulate *trans*-synaptic homophilic or heterophilic interactions through splicing-dependent, large conformational changes (*Berns et al., 2018*; *Gogou et al., 2024*; *Li et al., 2020*). While studying these cell adhesion molecules has highlighted how microexons can trigger differential protein–protein interactions, little is known about the developmental consequences of these biochemical switches in other protein classes.

Identifying the neurobiological roles of many individual microexons is a challenge. Studies of splicing regulators such as *srrm4* impact the entire splicing program, making it impossible to determine the importance of individual microexons to protein function. Furthermore, microexons could still be differentially included in a *srrm4* regulatory mutant via compensation by other splicing factors, such as its paralogue *srrm3*, which is responsible for microexon inclusion in photoreceptor transcripts in zebrafish (*Ciampi et al., 2022*). A vertebrate model is necessary for these studies; although there is a similar splicing mechanism in *Drosophila melanogaster*, there is no overlap of the individual microexons (*Torres-Méndez et al., 2021*). Thus, we used larval zebrafish, a vertebrate that supports more high-throughput studies than mammalian models (*Thyme et al., 2019*), to assess the brain activity, brain structure, and behavior of zebrafish larvae mutant for 45 microexons (*Figure 1A*).

## Results

A recently developed computational tool identified microexons that are differentially spliced during mouse brain development and conserved in zebrafish (*Parada et al., 2021*). Based on the intersection of these two sets, 95 microexons fulfilled both parameters (*Supplementary file 1*). Of these 95 microexons, 42 exist in a canonical layout in the zebrafish genome, with both a UGC and UC repeat – or similar polypyrimidine tract – directly upstream of the alternatively spliced exon (*Gonatopoulos-Pournatzis et al., 2018*; *Supplementary file 1*, *Figure 1—figure supplement 1*), indicating that Srrm4 likely controls their inclusion. Of the remaining microexons, 44 are organized similarly to the canonical layout, typically with either a UGC or UC repeat. Thus, they may also be regulated by Srrm4. The protein sequences of the 95 microexons are highly conserved between mouse and zebrafish

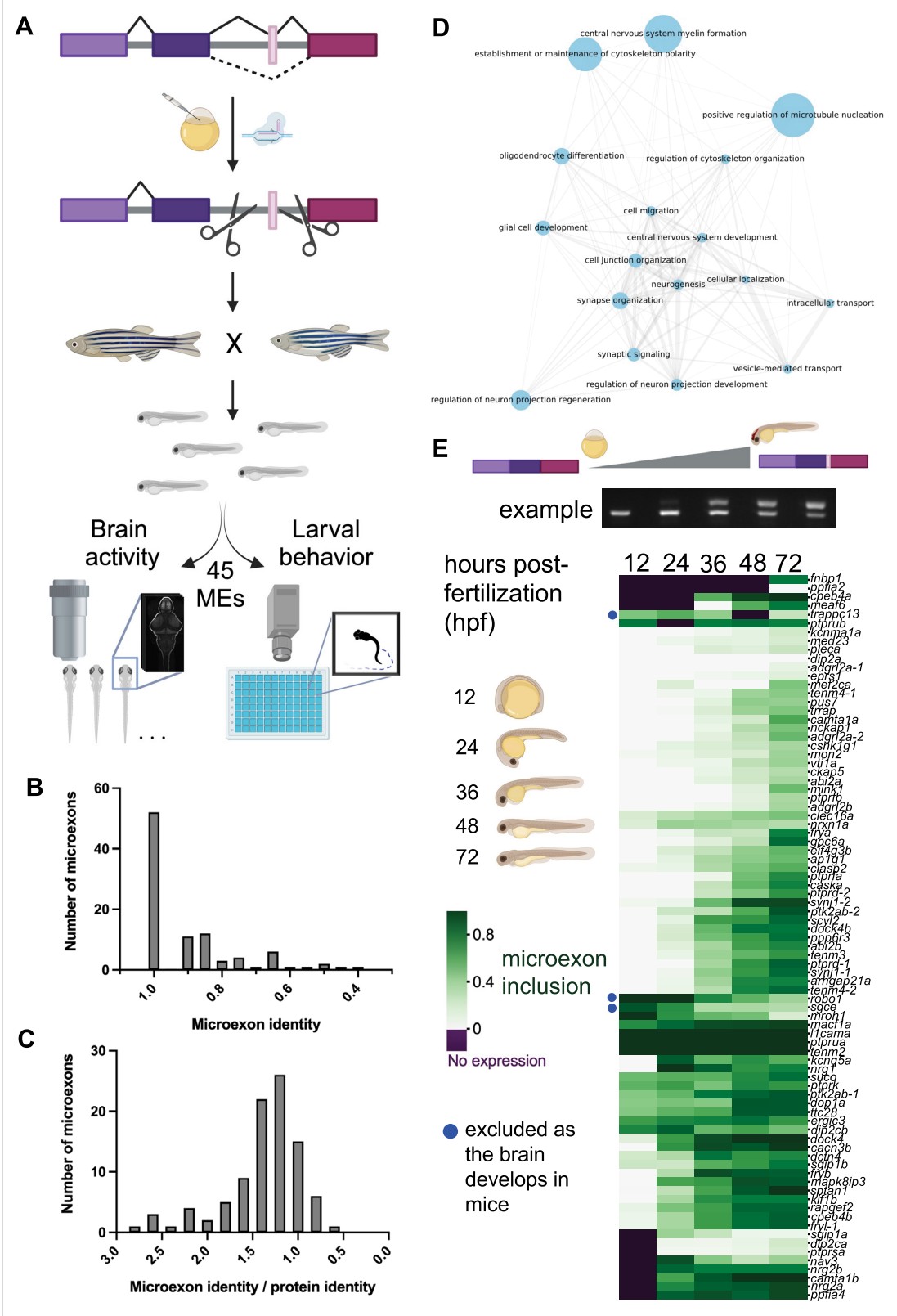

**Figure 1.** Generation and analysis of zebrafish with mutations that remove conserved, developmentally regulated microexons. (**A**) Pipeline of the screen. Mutant lines with alternatively spliced microexons removed were generated with CRISPR/Cas9, crossed together, and sibling larvae were assessed for changes to brain morphology, brain activity, and behavioral profiling. (**B**) The amino acid sequence identity of 95 zebrafish microexons compared to mouse. (**C**) The amino acid sequence identity of 95 zebrafish microexons compared to mouse and divided by the sequence identity of the entire

*Figure 1 continued on next page*

Figure 1 continued

protein. (**D**) Gene Ontology analysis of biological processes associated with the 95 mouse microexons that are conserved in zebrafish. The analysis was completed using the PANTHER classification system. (**E**) Quantification of reverse transcription PCR (RT-PCR) for microexon-containing regions over zebrafish development. These data were clustered using the default seaborn clustermap settings (method = 'average'). Panels A and E were created in BioRender.

The online version of this article includes the following source data and figure supplement(s) for figure 1:

**Source data 1.** Quantified reverse transcription PCR (RT-PCR) for microexon-containing regions over zebrafish development for panel E.

**Figure supplement 1.** Microexons in relationship to cut sites, neighboring exons, putative regulatory elements, and gRNA sites.

**Figure supplement 2.** Transcriptomic outcomes of eliminating selected microexons.

**Figure supplement 2—source data 1.** qRT-PCR for selected microexon lines normalized to *rpl13a*.

**Figure supplement 3.** Phenotypes for mutants in microexon genes that remove protein sequence beyond the microexon.

**Figure supplement 4.** Behavior summary data for mutants in microexon genes that remove protein sequence beyond the microexon.

(*Figure 1B*), more so than the full-length protein sequences (*Figure 1C*). The genes containing microexons are enriched for those involved in neuronal development as well as the cytoskeleton (*Figure 1D*). The majority of these microexons are included in transcripts as the brain develops, rather than lost, in both mouse and zebrafish (*Figure 1E*), making CRISPR/Cas9 removal of the microexon an ideal approach for studying the function of the added amino acids.

Using CRISPR/Cas9, we generated lines that removed 45 conserved microexons (*Supplementary file 2*) and assayed larval brain activity, brain structure, and behavior (*Figure 1A*). Four guide RNAs were used, two on each side of the microexon (*Supplementary file 2*, *Figure 1—figure supplement 1*). For microexons with upstream regulatory elements that are likely important for splicing, these elements were also removed (*Figure 1—figure supplement 1*). While we were most interested in those microexons only present after 24 hr post-fertilization (*Figure 1E*), as this is when the brain is forming, we also removed microexons from several genes with an earlier expression of the isoform. For eight mutant lines, we confirmed that the microexon was eliminated from the transcripts as expected (*Figure 1—figure supplement 2*). Although our genomic deletion did not yield unexpected isoforms, qRT-PCR on these eight lines revealed significant downregulation for the homozygous *vav2* mutant (*Figure 1—figure supplement 2*), indicating possibly complex genetic regulation. Protein-truncating mutations in eleven additional genes that contain microexons revealed developmental and neural phenotypes in zebrafish (*Figure 1—figure supplement 3*, *Figure 1—figure supplement 4*), indicating that the genes themselves are involved in biologically relevant pathways. Three of these genes – *tenm4*, *sptan1*, and *ppp6r3* – are also in our microexon line collection.

To uncover neural phenotypes, we assessed baseline and stimulus-driven movement. Homozygous mutant and control sibling larvae were subjected to a behavioral pipeline in 96-well plates from 4 to 7 days post-fertilization (dpf) (*Figure 2A*). Only a small number of strong, repeatable phenotypes were observed either at baseline or in response to acoustic or visual stimuli (*Figure 2B, C*, *Figure 2—figure supplement 1*, *Figure 2—figure supplement 2*, *Figure 2—figure supplement 3*). Baseline behavioral differences were observed in multiple movement categories. For example, *rapgef2* mutants showed an increased movement frequency (*Figure 2B, D*) but no changes to the characteristics of these movements (magnitude) or location in the well. In contrast, *dop1a* mutants had reduced daytime movement and an increased well-edge preference specifically at night (*Figure 2—figure supplement 2*, *Figure 2—figure supplement 4*, *Figure 2B, D*). Other baseline movement phenotypes included a well-center preference for *vav2* mutants, a decreased movement magnitude for *ptprd*-2 (also referred to in the literature as meA) mutants (*Figure 2—figure supplement 4*), and an increased movement magnitude for *ptprd*-1 (also referred to in the literature as meB) mutants (*Figure 2D*). Dark flash response phenotypes included a reduced latency for *eif4g3b* mutants and a reduced response frequency for *ptprd*-1 (meB) and *dop1a* mutants (*Figure 2C, D*, *Figure 2—figure supplement 4*). Acoustic response phenotypes included an increased frequency of response to strong stimuli for *ppp6r3* mutants and a reduced frequency of response to strong acoustic stimuli for *ptprd*-2.

To detect changes to brain development and function, we collected phospho-Erk (pErk) brain activity maps under unstimulated conditions at 6 dpf. Large morphological differences were identified using the Jacobian matrix derived from the image registration process (*Jefferis et al., 2007*; *Rohlfing and Maurer, 2003*). Similar to the behavioral findings, few phenotypes were observed in brain activity

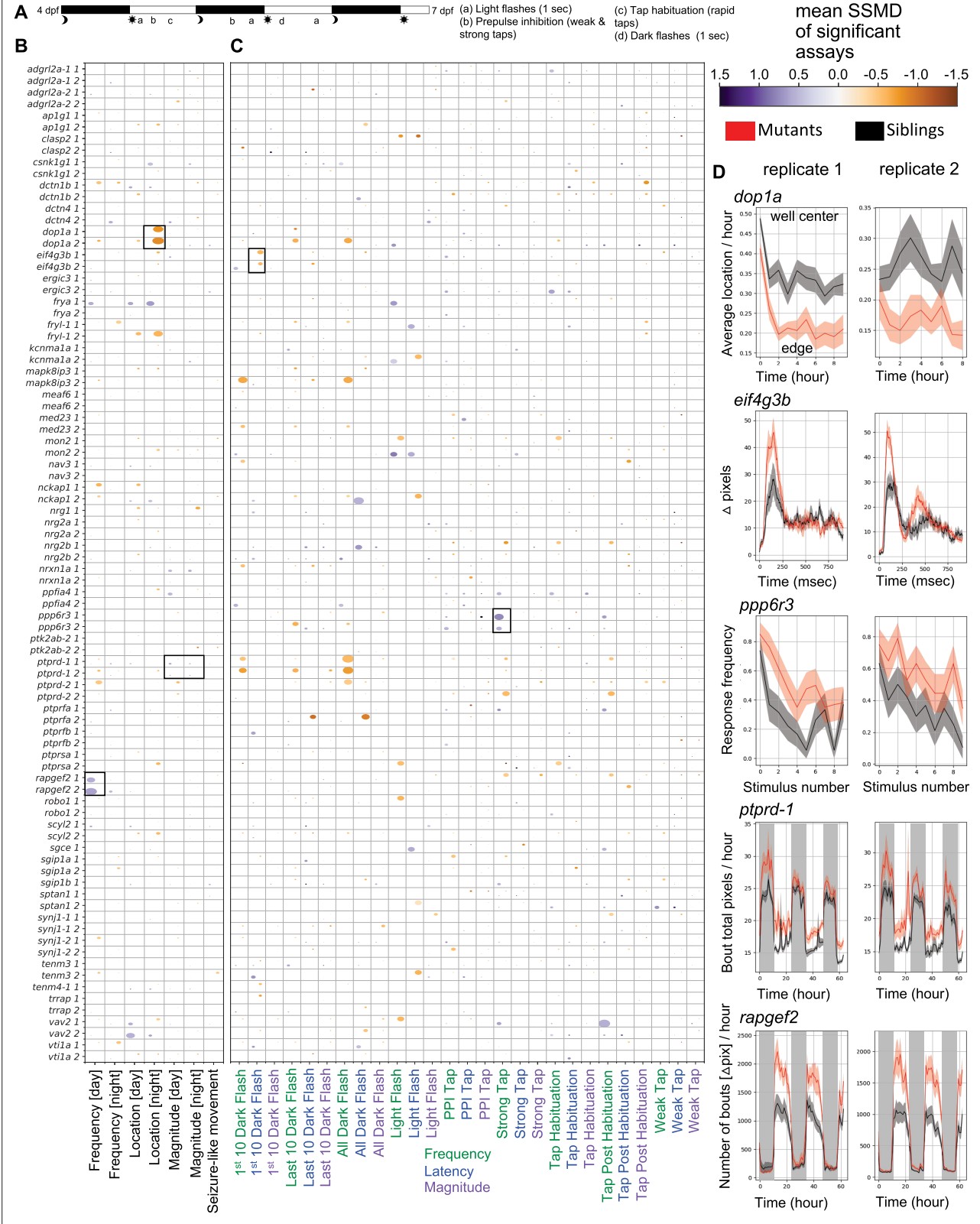

**Figure 2.** Larval behavioral phenotypes of zebrafish with microexons removed. (**A**) Summary of behavioral pipeline. (**B**) Baseline behavioral phenotypes for microexon mutants. The labels '1' and '2' indicate biological replicates. The data shown is for homozygous mutant larvae compared to wild-type siblings. Comparisons to the heterozygous larvae are removed for clarity and available in the Supplementary Materials, as they often have even milder phenotypes than homozygous. The size of the bubble represents the percent of significant measurements in the summarized category, and the color

*Figure 2 continued on next page*

*Figure 2 continued*

represents the mean of the strictly standardized mean difference (SSMD) of the significant assays in that category. (**C**) Stimulus-driven behavioral phenotypes for microexon mutants. The labels '1' and '2' indicate biological replicates. The bubble size and color are calculated the same as in panel B. (**D**) Examples of behavioral phenotypes. The black boxes in panels B and C correspond to the selected plots. Wild-type siblings (black) are compared to the homozygous (red), and the plots show mean ± SEM. All *N* are in **Supplementary file 2**. Kruskal–Wallis ANOVA p-values for the selected plots are as follows: *dop1a* center preference during the first night (day0night_boutcenterfraction_3600) = 0.00006/0.0008, *N* +/+ = 15 (run 1) and 14 (run 2), *N* −/− = 17 (run 1) and 15 (run 2); *eif4g3b* p-values are not calculated for response traces (shown is all dark flashes in block 1), p-values for the latency for the first 10 dark flashes in block 1 (day6dpfdf1a_responselatency) = 0.026/0.0008, *N* +/+ = 16 (run 1) and 15 (run 2), *N* −/− = 15 (run 1) and 21 (run 2); *ppp6r3* frequency of response to strong acoustic stimuli with a sound frequency of 1000 Hz that precede the habituation block (day5dpfhab1pre_responsefrequency_1_a1f1000d5p) = 0.002/0.016, *N* +/+ = 19 (run 1) and 20 (run 2), *N* −/− = 21 (run 1) and 29 (run 2); *ptprd*-1 pixels moved in each bout for the duration of the experiment (combo_boutcumulativemovement_3600) = 0.001/0.00002, *N* +/+ = 22 (run 1) and 21 (run 2), *N* −/− = 26 (run 1) and 18 (run 2); *rapgef2* number of bouts for the duration of the experiment calculated using the delta pixel data in each frame (dpix_numberofbouts_3600) = 0.002/0.001, *N* +/+ = 21 (run 1) and 23 (run 2), *N* −/− = 21 (run 1) and 20 (run 2).

The online version of this article includes the following source data and figure supplement(s) for figure 2:

**Source data 1.** Zip file of summarized behavior data and script to generate the heatmap in panel B.

**Source data 2.** Zip file of summarized behavior data and script to generate the heatmap in panel B.

**Figure supplement 1.** Baseline behavior summary data for microexon mutants.

**Figure supplement 2.** Frequency of motion example plots for microexon mutants.

**Figure supplement 3.** Stimulus-driven behavior summary data for microexon mutants.

**Figure supplement 4.** Additional behavioral phenotypes for microexon mutants.

(*Figure 3*, *Figure 3—figure supplement 1A*, *Figure 3—figure supplement 2*) or brain structure (*Figure 3B*, *Figure 3—figure supplement 3*, *Figure 3—figure supplement 4*) in homozygous mutants compared to wild-type control siblings when quantified by region (*Figure 3C*). Repeatable phenotypes were mild (*Figure 3D*) compared to a mutant with a moderate phenotype and both increased and decreased activity and structure (see control-*tcf7l2*) (*Capps et al., 2024*). Only one mutant, *ppp6r3*, displayed substantial and repeatable differences in brain morphology (*Figure 3B, E*), with reduced size observed mainly in the thalamus and neighboring pretectum. While the loss of both copies of microexons was required for most phenotypes, brain activity differences were observed in *vti1a* and *sptan1* heterozygotes (*Figure 3F*) as well as *vav2* and *mapk8ip3* (*Figure 3—figure supplement 1*, *Figure 3—figure supplement 2*). While repeatable phenotypes in brain activity and structure were observed in a small subset of mutants, other zebrafish studies using identical methods have typically uncovered brain activity phenotypes for approximately half of the lines that are stronger than any we observed (*Figure 3G*; *Capps et al., 2024*; *Thyme et al., 2019*; *Weinschutz Mendes et al., 2023*).

## Discussion

Our study provides a broad overview of the larval zebrafish neural phenotypes resulting from removing developmentally regulated microexons. Here, we leveraged our previously successful brain activity mapping and behavioral profiling strategy, which has high sensitivity and has uncovered phenotypes for dozens of mutants (*Thyme et al., 2019*). While only a few zebrafish lines had neural phenotypes, most of the microexons with behavioral or brain activity differences were previously unstudied and found in diverse protein classes.

The small number of observed phenotypes was unexpected based on conservation of the microexons (*Figure 1B, C*) and the discovery rates of previous screens using similar methods (*Capps et al., 2024*; *Thyme et al., 2019*; *Weinschutz Mendes et al., 2023*). Although most of these studies focused on protein-truncating alleles, we have found phenotypes in lines with missense mutations (*Capps et al., 2024*). However, this outcome is consistent with a recent, similar study of microexons in zebrafish (*Lopez-Blanch et al., 2024*). Although only three of the 18 microexons this group studied are in the same genes as our set (*ap1g1*, *vav2*, and *vti1a*), the phenotypes for individual mutants are similar, and this work shares the same overall finding of minimal phenotypes in zebrafish larvae. Importantly, the pErk brain activity mapping method we used is highly sensitive, significantly minimizing the likelihood that substantial zebrafish larval phenotypes from individual microexon removal are being missed. In our published work (*Capps et al., 2024*; *Thyme et al., 2019*), we showed that brain activity can be drastically impacted without manifesting in differences in those behaviors assessed in a typical larval

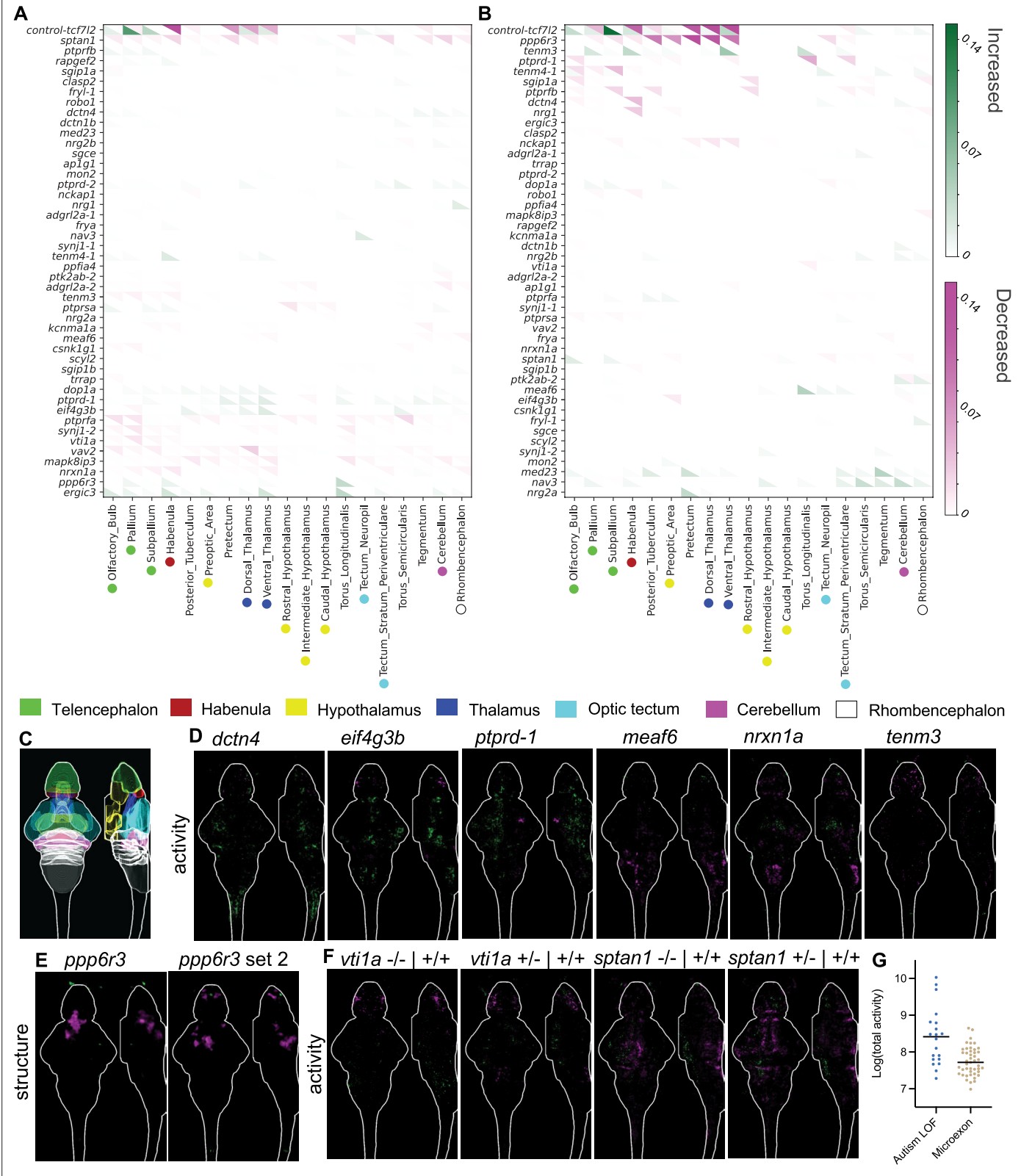

**Figure 3.** Whole-brain activity and morphology phenotypes of zebrafish with microexons removed. (**A**) Clustered summary of pErk comparisons between homozygous mutants and wild-type control siblings, where magenta represents decreased activity and green represents increased. The signal in each region was summed and divided by the region size. The *N* for all experiments is available in ***Supplementary file 2***. (**B**) Clustered summary of structure comparisons between homozygous mutants and wild-type control siblings, where magenta represents decreased size and green represents

*Figure 3 continued*

increased. (**C**) Location of major regions in the zebrafish brain based on masks from the Z-Brain atlas (*Randlett et al., 2015*). (**D**) Brain imaging for microexon mutants with repeatable brain activity phenotypes. The brain images represent the significant signal difference between homozygous and wild-type control siblings. They are shown as sum-of-slices projections (*Z*- and *X*-axes) with the white outline representing the zebrafish brain. (**E**) Structural phenotype of the *ppp6r3* mutant, with replicates shown side-by-side. The magenta indicates decreased size. (**F**) Brain imaging for two microexon mutants with brain activity phenotypes that are similar for both the heterozygous and homozygous mutants. (**G**) Comparison of the total brain activity signal between homozygous microexon mutants from this work and mutants for autism risk genes from *Capps et al., 2024*. Both increased and decreased activity are considered a single comparison rather than the average of the repeats.

The online version of this article includes the following source data, source code, and figure supplement(s) for figure 3:

**Source code 1.** Python script that sums the intensities from image stacks to generate the source data for panels A and B.

**Source code 2.** Python script that generates the heatmap in panels A and B using the source data.

**Source data 1.** Red channel signal for activity differences heatmap in panel A.

**Source data 2.** Green channel signal for structural differences heatmap in panel A.

**Source data 3.** Red channel signal for structural differences heatmap in panel B.

**Source data 4.** Green channel signal for structural differences heatmap in panel B.

**Figure supplement 1.** Brain activity maps for multiple genetic comparisons for microexon mutants.

**Figure supplement 2.** Second set of brain activity maps for multiple genetic comparisons for microexon mutants.

**Figure supplement 3.** Brain structural maps for multiple genetic comparisons for microexon mutants.

**Figure supplement 4.** Second set of brain structural maps for multiple genetic comparisons for microexon mutants.

screen such as was completed by Lopez-Blanch et al. Furthermore, the genes that contain microexons are developmentally important (*Figure 1—figure supplement 3*, *Figure 1—figure supplement 4*). For instance, the *caska* loss-of-function mutant displayed substantial changes to brain activity, a smaller brain size, and multiple behavioral phenotypes. Loss-of-function mutants for *sptan1* and *ppp6r3* are stronger than lines that only remove the microexon (*Figure 1—figure supplement 3*, *Figure 1—figure supplement 4*).

The limited effects of individual microexon deletions suggest that stronger phenotypes may require disrupting multiple microexons or examining later stages of development. It is possible that perturbation of multiple microexons simultaneously, as was observed in the brains of autistic individuals (*Irimia et al., 2014*), is necessary to produce pronounced effects on early neurodevelopment. Zebrafish homozygous for *srrm4* deletion did not display any overt developmental phenotypes (*Ciampi et al., 2022*), in contrast to a previous morpholino study (*Calarco et al., 2009*). More recently, it was revealed that loss of *srrm4* has minimal impacts on behavior (*Lopez-Blanch et al., 2024*) and brain structure (*Gupta et al., 2025*), while *srrm3* mutants have significant neural phenotypes and early mortality (*Ciampi et al., 2022*; *Lopez-Blanch et al., 2024*). This finding indicates that Srrm3 loss is necessary for complete elimination of these microexons, but studying microexons in the context of this line is also complicated by possible downstream consequences of severe vision loss. Alternatively, although the microexons are present at larval stages, they may only become important during later stages of neuronal maturation that support more complex behaviors. Mouse models were studied as adults and for disruptions to social interaction, learning, and memory (*Gonatopoulos-Pournatzis et al., 2020*; *Han et al., 2024*), which larvae are not yet capable of *Dreosti et al., 2015*; *Valente et al., 2012*.

Our study extends on findings for previously studied genes containing microexons. We discovered increased brain activity (*Figure 3D*) and an altered dark flash response in mutants for the *eif4g3b* (*Figure 2D*), whereas a mouse model with the microexon removed was described as having no behavior phenotypes (*Gonatopoulos-Pournatzis et al., 2020*). That study, however, focused on social behavior, learning, and memory and may not have included stimuli analogous to dark flashes. Alternatively, microexon removal in zebrafish could have differential behavioral impacts than in rodents. Mice mutant for the *ptprd*-2 (meA) microexon have increased movement and interrupted sleep (*Park et al., 2020*). Both zebrafish mutants in *ptprd* demonstrate the opposite, with reduced daytime movement and increased sleep during the day for *ptprd*-2 (*Figure 2—figure supplement 4*). While the underlying disrupted circuitry could differ between the species, the daytime sleep increase raises the possibility of a rebound response to undetected nighttime sleep disruptions. In other cases, the

gene that contains the microexon has been extensively studied, but research on the microexon itself is absent. For example, the function of SNARE protein Vti1a in neural development and dense core vesicle biogenesis has been studied in great depth (*Bollmann et al., 2022*; *Emperador-Melero et al., 2018*; *Kunwar et al., 2011*; *Sokpor et al., 2021*; *Walter et al., 2014*). Its brain-specific microexon, however, was recognized over 20 years ago (*Antonin et al., 2000*), and its specific importance has been largely ignored. We discovered reduced brain activity mainly in the telencephalon in both homozygous and heterozygous lines compared to wild-type siblings (*Figure 3F*), highlighting the relevance of considering multiple conserved isoforms when characterizing protein function in the brain.

Mutants for unstudied microexons with the strongest phenotypes come from diverse protein classes (*Gonatopoulos-Pournatzis and Blencowe, 2020*), indicating this splicing program impacts neurodevelopmental processes beyond *trans*-synaptic partner selection. The only mutant with a substantial brain size difference was *ppp6r3*, which encodes a regulator of protein phosphatase 6 (PP6), and this reduced size was present and more severe in the predicted protein-truncating *ppp6r3$^{sa16892}$* line (*Figure 1—figure supplement 3*). Although little is known about this protein, it has been linked to cell cycle (*Heo et al., 2020*) and cell death (*Wu et al., 2022*), nominating hypotheses for determining future research on the source of the reduced size. Dctn4 is part of the dynactin complex, Mapk8ip3 is involved in Kinesin-1-Dependent axonal trafficking (*Platzer et al., 2019*; *Watt et al., 2015*), and Sptan1 is a cytoskeleton scaffold protein (*Huang et al., 2017*). The number of microexon-containing genes related to the cytoskeleton (*Figure 1D*) and also trafficking (e.g., *vti1a*) suggests that microexons may be important to the specialized cytoskeletal needs of developing neurons. The functions of microexons in cytoskeletal proteins are beginning to be discovered (*Poliński et al., 2023*). Transcriptional regulation is represented by the *meaf6* mutant, while Rapgef2, which has the strongest behavioral phenotypes upon microexon removal (*Figure 3D*), is a signaling protein.

Although our list is not comprehensive, as many microexons remain unstudied, we have prioritized several microexons for future study from the 45 tested. For seven mutants, we confirmed that the microexon was cleanly removed without unanticipated effects on isoforms or transcript levels (*Figure 1—figure supplement 2*), including for some of the most interesting that are entirely unstudied (*ppp6r3*, *sptan1*, *meaf6*). For those with phenotypes that are not yet confirmed by qRT-PCR (*rapgef2*, *dctn4*, *dop1a*, *mapk8ip3*), studies of gene loss-of-function in zebrafish and mice reveal far stronger phenotypes and lethality, suggesting that our lines impact only the microexon (*Abeler-Dörner et al., 2020*; *Drerup and Nechiporuk, 2013*; *Satyanarayana et al., 2010*; *Tuttle et al., 2019*). In future work, proximity-dependent ligation in zebrafish lines with and without the microexon could reveal differential binding partners in vivo (*Rosenthal et al., 2021*), as microexons are often found in surface-accessible protein domains and can regulate protein–protein interactions (*Dergai et al., 2010*; *Irimia et al., 2014*). The discovery of microexon mutants with neural phenotypes lays the groundwork for future investigation of impacted neurodevelopmental pathways and how interactomes differ between isoforms.

## Materials and methods

### Key resources table

| Reagent type (species) or resource | Designation | Source or reference | Identifiers | Additional information |
|---|---|---|---|---|
| Genetic reagent (*D. rerio*) | Microexon mutants | This paper | *Supplementary file 2* | |
| Antibody | Mouse monoclonal anti-Erk | Cell Signaling | #4696 | IF(3:1000) |
| Antibody | Rabbit monoclonal anti-phospho-Erk | Cell Signaling | #4370 | IF(1:500) |
| Commercial assay or kit | E.Z.N.A. MicroElute Total RNA Kit | Omega Bio-Tek | R6834-02 | |
| Commercial assay or kit | iScript Reverse Transcription Supermix | Bio-Rad | #1708840 | |
| Commercial assay or kit | GoTaq 2x master mix | Promega | M7123 | |
| Commercial assay or kit | SsoAdvanced Universal SYBR Green Supermix | Bio-Rad | #1725270EDU | |

*Continued on next page*

*Continued*

| Reagent type (species) or resource | Designation | Source or reference | Identifiers | Additional information |
|---|---|---|---|---|
| Sequence-based reagent | Oligonucleotide primers for genotyping and cDNA amplification | Life technologies, this paper | *Supplementary files 1 and 2* | |
| Software, algorithm | Fiji/ImageJ | https://imagej.net/software/fiji/downloads; *Schindelin et al., 2012* | | |
| Software, algorithm | Image registration with CMTK | https://www.nitrc.org/projects/cmtk; *Jefferis et al., 2007* | | |
| Software, algorithm | Zebrafish brain mapping and Z-Brain atlas | *Randlett et al., 2015* | | |
| Software, algorithm | Zebrafish behavior analysis | https://github.com/thymelab/ZebrafishBehavior; *Joo et al., 2020* | | |

## Zebrafish husbandry

Zebrafish experiments were approved by the UAB Institutional Animal Care and Use Committee (IACUC protocols 22155 and 21744) and UMass Chan Institutional Animal Care and Use Committee (IACUC protocol 202300000053). Mutants were generated in an Ekkwill-based strain using CRISPR/Cas9 as previously described (*Capps et al., 2024*; *Thyme et al., 2019*; *Supplementary file 2*). Both adults and larvae were maintained on a 14/10 hr light/dark cycle at 28°C. Experimental larvae were maintained at a density of less than 160 per dish in 150 mm Petri dishes in fish water with methylene blue, and debris was removed at least twice prior to experimentation. Visibly unhealthy larvae and those without inflated swim bladders were excluded from experiments. Control animals were always siblings from the same clutch and derived from a single parental pair, and all larvae were genotyped after experimentation. Even with using sibling controls and collecting multiple biological replicates from individual parents, the possibility remains that linked genetic variation may have contributed to the mild phenotypes we observed, as only a single line was generated.

## Brain activity and morphology

Phosphorylated-ERK (pErk) antibody staining was conducted as previously described (*Capps et al., 2024*; *Thyme et al., 2019*). The total ERK antibody (Cell Signaling, #4696) was used at 3:1000 and pErk antibody was used at 1:500 (Cell Signaling, #4370). Typically, the primary antibody exposure was for 2–3 days and secondary for 1 day. Image stacks were collected with a Zeiss LSM 900 upright confocal microscope using a 20x/1.0 NA water-dipping objective. These stacks were registered to the standard zebrafish Z-Brain reference using Computational Morphometry Toolkit (CMTK) (*Jefferis et al., 2007*; *Randlett et al., 2015*; *Rohlfing and Maurer, 2003*; *Thyme et al., 2019*). The significance threshold for MapMAPPING (*Randlett et al., 2015*; *Thyme et al., 2019*) was set based on a false discovery rate where 0.05% of control pixels would be called as significant for both the brain activity and structural measurements.

## Larval behavior

Larval behavior assays were conducted and analyzed as previously described (*Capps et al., 2024*; *Joo et al., 2020*; *Thyme et al., 2019*). The pipeline begins on the evening of larval stage 4 dpf and includes the following stimuli: 5 dpf light flashes (9:11 to 9:25), mixed acoustic stimuli (prepulse, strong, and weak, from 9:38 to 2:59), and three blocks of acoustic habituation (3:35 to 6:35); the night between 5 and 6 dpf mixed acoustic stimuli (1:02 to 5:00) and light flashes (6:01 to 6:20); 6 dpf three blocks of dark flashes with an hour between them (10:00 to 3:00) and mixed acoustic stimuli and light flashes (4:02 to 6:00). Stimulus responses were quantified from high-speed (285 fps) 1-s-long movies. All code for behavior analysis is available at https://github.com/thymelab/ZebrafishBehavior (copy archived at *Thyme, 2024a*). Frequency of movement, magnitude of movement, and location preferences were calculated for the baseline data. An example of a 'frequency' measure would be the active seconds/hour. An example of the 'magnitude' measure would be the movement velocity of a bout. An example of a 'location' measure would be the fraction of the bout time spent in the center of the well. Frequency, response magnitude parameters, and latency to response are also calculated for each stimulus response from the high-speed movies. The stimulus responses are also broken up

into subsets, such as the dark flashes that occur at the beginning of the dark flash block and those that occur at the end. Multiple measures are calculated for the experiment, and both summarized and raw measures are shown in *Figure 2* and the Supplementary Material. The summarized data is generated by merging related terms (e.g., all the 'frequency' measures) together. The size of the bubble represents the percent of significant measurements in the summarized category, and the color represents the mean of the strictly standardized mean difference of the significant assays in that category. It is possible for some measures in the merged group to be increased (purple) and some decreased (orange), such as if a behavioral change occurs between 4 and 6 dpf, which is why it is possible to have two offset bubbles in each square. The scripts for merging these measurements and generating the summarized bubble plot graphs are available at https://github.com/thymelab/DownstreamAnalysis (copy archived at *Thyme, 2024b*).

## Reverse transcription PCR (RT-PCR and qRT-PCR)

RNA was extracted from zebrafish embryos using the E.Z.N.A. MicroElute Total RNA Kit (Omega Bio-Tek R6834-02), and cDNA synthesis was performed with iScript Reverse Transcription Supermix (Bio-Rad #1708840). Following PCR with GoTaq polymerase (Promega M7123) and primers (*Supplementary file 1*), samples were run on a 4% agarose gel and the band intensities were quantified using Fiji (*Schindelin et al., 2012*). For qRT-PCR, we combined three to four zebrafish heads at 6 dpf for each biological replicate. For RT-PCR, these separate biological samples were combined into one per genotype. We homogenized the heads and performed RNA extraction and cDNA synthesis as described above. We then performed qRT-PCR with SsoAdvanced Universal SYBR Green Supermix (Bio-Rad #1725270EDU) according to the manufacturer's protocol and using primers in *Supplementary file 1*.

## Acknowledgements

This research was supported by a Mallinckrodt Award from the Edward Mallinckrodt Jr. Foundation (SBT). Guillermo Parada was instrumental to this work by sharing his list of conserved and developmentally regulated microexons while his study was still a preprint under review. We thank the UAB fish facility staff, UMass Chan fish facility staff, the Zebrafish International Resource Center (ZIRC), particularly Andrzej Nasiadka, the Research Computing team at UAB, and the UAB Department of Neurobiology for supporting this study. We also thank the following Thyme lab members for experimental assistance or training that supported this study: Brandon Bastein, Anna Moyer, Lynne Zhou, Vaishnavi Balaji, Alexia Barcus, and Sahil Malhotra.

## Additional information

### Competing interests

Summer B Thyme: Reviewing editor, eLife. The other authors declare that no competing interests exist.

### Funding

| Funder | Grant reference number | Author |
| --- | --- | --- |
| Edward Mallinckrodt, Jr. Foundation | | Summer B Thyme |

The funders had no role in study design, data collection, and interpretation, or the decision to submit the work for publication.

### Author contributions

Caleb CS Calhoun, Investigation, CCSC collected most of the experimental data for the original manuscript draft; Mary ES Capps, Formal analysis, Investigation, Writing – review and editing, MESC completed all revision experiments and analysis and contributed to the experimental work in the original draft; Kristie Muya, William C Gannaway, Claire L Conklin, Morgan C Klein, Emma G Torija-Olson,

Investigation; Verdion Martina, Investigation, VM generated the mutant lines alongside JWM; Jhodi M Webster, Investigation, JWM generated the mutant lines alongside VM; Summer B Thyme, Conceptualization, Resources, Data curation, Software, Formal analysis, Supervision, Funding acquisition, Validation, Visualization, Writing – original draft, Project administration, Writing – review and editing

### Author ORCIDs
Emma G Torija-Olson ⓘ https://orcid.org/0000-0001-5003-7395
Summer B Thyme ⓘ https://orcid.org/0000-0003-3593-4148

### Ethics
Zebrafish experiments were approved by the UAB Institutional Animal Care and Use Committee (IACUC protocols 22155 and 21744) and UMass Chan Institutional Animal Care and Use Committee (IACUC protocol 202300000053).

Reviewer #1 (Public review): https://doi.org/10.7554/eLife.101790.3.sa1
Reviewer #3 (Public review): https://doi.org/10.7554/eLife.101790.3.sa2
Author response https://doi.org/10.7554/eLife.101790.3.sa3

## Additional files

### Supplementary files
Supplementary file 1. Sequences and locations of 95 microexons conserved between zebrafish and mouse.

Supplementary file 2. Mutants generated and corresponding genotyping and experimental information.

MDAR checklist

### Data availability
The mutants described in the paper are available from ZIRC. Code is available from https://github.com/thymelab/ZebrafishBehavior (copy archived at *Thyme, 2024a*) and https://github.com/thymelab/DownstreamAnalysis (copy archived at *Thyme, 2024b*). Processed behavioral and imaging data is available from Zenodo under the following DOIs: https://doi.org/10.5281/zenodo.13137486 (behavior output files), https://doi.org/10.5281/zenodo.13138646 (behavior input files), and https://doi.org/10.5281/zenodo.13138766 (brain activity mapping stacks).

The following datasets were generated:

| Author(s) | Year | Dataset title | Dataset URL | Database and Identifier |
|---|---|---|---|---|
| Thyme SB | 2024 | Removal of developmentally regulated microexons has a minimal impact on larval zebrafish brain morphology and function - behavior data output | https://doi.org/10.5281/zenodo.13137486 | Zenodo, 10.5281/zenodo.13137486 |
| Thyme SB | 2024 | Removal of developmentally regulated microexons has a minimal impact on larval zebrafish brain morphology and function - behavior data input files | https://doi.org/10.5281/zenodo.13138646 | Zenodo, 10.5281/zenodo.13138646 |
| Thyme SB | 2024 | Removal of developmentally regulated microexons has a minimal impact on larval zebrafish brain morphology and function - imaging stacks | https://doi.org/10.5281/zenodo.13138766 | Zenodo, 10.5281/zenodo.13138766 |

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
