## [Editor Report · eLife Assessment]

This **important** work examines how microexons contribute to brain activity, structure, and behavior. The authors find that loss of microexon sequences generally has subtle impacts on these metrics in larval zebrafish, with few exceptions. The evidence is **solid**, using modern high-throughput phenotyping methodology in zebrafish. Overall, this work will be of interest to neuroscientists and generate further studies of interest to the field.

---

## [Referee Report · Reviewer #1 (Public review)]

Summary:

The authors use high-throughput gene editing technology in larval zebrafish to address whether microexons play important roles in the development and functional output of larval circuits. They find that individual microexon deletions rarely impact behavior, brain morphology, or activity, and raise the possibility that behavioral dysregulation occurs only with more global loss of microexon splicing regulation. Other possibilities exist: perhaps microexon splicing is more critical for later stages of brain development, perhaps microexon splicing is more critical in mammals, or perhaps the behavioral phenotypes observed when microexon splicing is lost are associated with loss of splicing in only a few genes.

Strengths:

- The authors provide a qualitative analysis of microexon inclusion during early zebrafish development

- The authors provide comprehensive phenotyping of microexon mutants, addressing the role of individual microexons in the regulation of brain morphology, activity, and behavior.

---

## [Referee Report · Reviewer #3 (Public review)]

Summary:

This paper sought to understand how microexons influence early brain function. By selectively deleting a large number of conserved microexons and then phenotyping the mutants with a behavior and brain activity assays, the authors find that most microexons have minimal effects on the global brain activity and broad behaviors of the larval fish-- although a few do have phenotypes.

Strengths:

The work takes full advantage of the scale that is afforded in zebrafish, generating a large mutant collection that is missing microexons and systematically phenotyping them with high throughput behaviour and brain activity assays. The work lays an important foundation for future studies that seek to uncover the likely subtle roles that single microexons will play in shaping development and behavior.

Weaknesses:

Although the manuscript includes evidence for many mutants that microexon deletion has minimal effect on full length transcript levels, some of the microexon loss does alter transcript levels. Since the mutations usually yielded no phenotype, these effects on full-length transcripts are unlikely to be a major confound. For mircoexon mutants displaying phenotypes, future work will have to tease apart whether secondary effects on the transcripts are contributing to the phenotype.

---

## [Author Response]

The following is the authors’ response to the original reviews.

**Reviewer #1 (Public review):**
Summary:The authors use high-throughput gene editing technology in larval zebrafish to address whether microexons play important roles in the development and functional output of larval circuits. They find that individual microexon deletions rarely impact behavior, brain morphology, or activity, and raise the possibility that behavioral dysregulation occurs only with more global loss of microexon splicing regulation. Other possibilities exist: perhaps microexon splicing is more critical for later stages of brain development, perhaps microexon splicing is more critical in mammals, or perhaps the behavioral phenotypes observed when microexon splicing is lost are associated with loss of splicing in only a few genes.A few questions remain:(1) What is the behavioral consequence for loss of srrm4 and/or loss-of-function mutations in other genes encoding microexon splicing machinery in zebrafish?

It has been established that srrm4 mutants exhibit no overt morphological phenotypes and are not visually impaired (Ciampi et al., 2022). We are coordinating our publication with Lopez-Blanch et al. (https://doi.org/10.1101/2024.10.23.619860), which shows that srrm4 mutants also have minimal behavioral phenotypes. In contrast, srrm3 mutants have severe vision loss, early mortality, and numerous neural and behavioral phenotypes (Ciampi et al., 2022; Lopez-Blanch et al., 2024). We now point out the phenotypes of srrm3/srrm4 mutants in the manuscript.

We chose not to generate and characterize the behavior and brain activity of srrm3/srrm4 mutants for two reasons: (1) we were aware of two other labs in the zebrafish community that had generated srrm3 and/or srrm4 mutants (Ciampi et al., 2022 and Gupta et al., 2024, https://doi.org/10.1101/2024.11.29.626094; Lopez-Blanch et al., 2024, https://doi.org/10.1101/2024.10.23.619860), and (2) we were far more interested in determining the importance of individual microexons to protein function, rather than loss of the entire splicing program. Microexon inclusion can be controlled by different splicing regulators, such as srrm3 (Ciampi et al., 2022) and possibly other unknown factors. Genetic compensation in srrm4 mutants could also result in microexons still being included through actions of other splicing regulators, complicating the analysis of these regulators. We mention srrm4 in the manuscript to point out that some selected microexons are adjacent to regulatory elements expected of this pathway. We did not, however, choose microexons to mutate based on whether they were regulated by Srrm4, making the characterization of srrm3/srrm4 mutants disconnected from our overarching project goal.

We have edited the Introduction as follows to clarify our goal: “Studies of splicing regulators such as srrm4 impact the entire splicing program, making it impossible to determine the importance of individual microexons to protein function. Further, microexons could still be differentially included in a regulatory mutant via compensation by other splicing factors ...”

(2) What is the consequence of loss-of-function in microexon splicing genes on splicing of the genes studied (especially those for which phenotypes were observed).

We are unclear whether “microexon splicing genes” refers to the splicing regulators srrm3/srrm4, which we choose not to study in this work (see response to point #1 above), or the genes that contain microexons. The severe visual phenotypes of srrm3 mutants confounds the study of microexon splicing in this line because altered splicing levels could be due to downstream changes in this significantly different developmental context. A detailed discussion of splicing consequences on removal of microexons from microexoncontaining genes is in the response to point #4 below.

(3) For the microexons whose loss is associated with substantial behavioral, morphological, or activity changes, are the same changes observed in loss-of-function mutants for these genes?

In the first version of the manuscript, we had included two explicit comparisons of microexon loss with a standard loss-of-function allele, one with a phenotype and one without, in Figure S1 (now Figures S3 and S4) of this manuscript. Beyond the two pairs we had included, Lopez-Blanch et al. (https://doi.org/10.1101/2024.10.23.619860) described mild behavioral phenotypes for a microexon removal for kif1b, and we showed developmental abnormalities for the kif1b loss-of-function allele (now Figure S3). We have now added a predicted protein-truncating allele for ppp6r3. This new line has phenotypes that are similar but slightly stronger in brain activity and structure than the mutant that lacks only the microexon. The prior Figure S1 (now Figures S3 and S4) was only briefly mentioned in the first version of the manuscript, and we now clarify this point in the Results: “Protein-truncating mutations in eleven additional genes that contain microexons revealed developmental and neural phenotypes in zebrafish (Figure S3, Figure S4), indicating that the genes themselves are involved in biologically relevant pathways. Three of these genes– tenm4, sptan1, and ppp6r3 – are also in our microexon line collection.”

Additionally, we can draw expected conclusions from the literature, as some genes with our microexon mutations have been studied as typical mutants in zebrafish or mice. We have modified our manuscript to include a discussion of both loss-of-function zebrafish and mouse mutants. See the response to below point #4.

(4) Do "microexon mutations" presented here result in the precise loss of those microexons from the mRNA sequence? E.g. are there other impacts on mRNA sequence or abundance?

We acknowledge that unexpected changes to the mRNA of the tested mutants could occur following microexon removal. In particular, all regulatory elements should be removed from the region surrounding the microexon, as any remaining elements could drive the inclusion of unexpected exons that result in premature stop codons.

First, we have clarified our generated mutant alleles by adding a figure (Figure S1) that details the location of the gRNA cut sites in relation to the microexon, its predicted regulatory elements, and its neighboring exons.

Second, we have experimentally determined whether the mRNA was modified as expected for a subset of mutants with phenotypes. In all eight tested lines (Figure S2), the microexon was precisely eliminated without causing any other effects on the sequence of the transcript in the neighboring region. We did, however, observe an effect on transcript abundance for one homozygous mutant (vav2). It is possible that complex forms of genetic regulation are occurring that are not induced by unexpected isoforms or premature stop codons. Interestingly, Lopez-Blanch et al. (https://doi.org/10.1101/2024.10.23.619860) eliminated a different microexon in vav2 and also observed a subtle well center preference. If their allele from an entirely different intronic region also results in transcript downregulation, it would support the hypothesis of genetic compensation through atypical pathways. If not, it is likely this phenotype is due specifically to removal of the microexon protein sequence. Not all mutants with phenotypes could be assessed with qRT-PCR because some were no longer present in the lab. All lines were generated in a similar way, however, removing both the microexon and neighboring regulatory elements while avoiding the neighboring exons. Accordingly, we now also explicitly point out those where the clean loss of the microexon was confirmed (eif4g3b, ppp6r3, sptan1, vti1a, meaf6, nrxn1a, tenm3) and those with possibly interesting phenotypes that were not confirmed (ptprd-1, ptprd-2, rapgef2, dctn4, dop1a, mapk8ip3).

Third, we have further emphasized in the manuscript that these observed phenotypes are extremely mild compared to those observed in over one hundred protein-truncating mutations we have assessed in previous (Thyme et al., 2019; Capps et al., 2024) and unpublished ongoing work. We showed data for one mutant, tcf7l2, which we consider to have moderately strong neural phenotypes, and we have extended this comparison in the revision (new Figure 3G). Additionally, loss-of-function alleles for some microexoncontaining genes have strong developmental phenotypes, as we showed in Figure S1 (now Figures S3 and S4) of this manuscript in addition to our published work (Thyme et al., 2019; Capps et al., 2024). It is known from the literature that the loss-of-function mutants for mapk8ip3 are stronger than we observed here (Tuttle., et al., 2019), suggesting that only the microexon is removed in our line. The microexons in Ptprd are also well-studied in mice, and we expect that only the microexon was removed in our lines. Both Dctn4 and Rapgef2 are completely lethal prior to weaning in mice (the International Mouse Phenotyping Consortium).

(5) Microexons with a "canonical layout" (containing TGC / UC repeats) were selected based on the likelihood that they are regulated by srrm4. Are there other parallel pathways important for regulating the inclusion of microexons? Is it possible to speculate on whether they might be more important in zebrafish or in the case of early brain development?

The microexons were not selected based on the likelihood that they were regulated by Srrm4. We have clarified the manuscript regarding this point. There are parallel pathways that can control the inclusion of microexons, such as Srrm3 (Ciampi et al., 2022). It is wellknown that loss of srrm3 has a stronger impact on zebrafish development than srrm4 (Ciampi et al., 2022). The goal of our work was not to investigate these splicing regulators but instead to determine the individual importance of these highly conserved protein changes.

Strengths:(1) The authors provide a qualitative analysis of splicing plasticity for microexons during early zebrafish development.(2) The authors provide comprehensive phenotyping of microexon mutants, addressing the role of individual microexons in the regulation of brain morphology, activity, and behavior.

We thank the reviewer for their support. The pErk brain activity mapping method is highly sensitive, significantly minimizing the likelihood that the field has simply not looked hard enough for a neural phenotype in these microexon mutants. In our published work (Thyme et al., 2019), we show that brain activity can be drastically impacted without manifesting in differences in those behaviors assessed in a typical larval screen (e.g., tcf4, cnnm2, and more).

Weaknesses:(1) It is difficult to interpret the largely negative findings reported in this paper without knowing how the loss of srrm4 affects brain activity, morphology, and behavior in zebrafish.

See response to point 1.

(2) The authors do not present experiments directly testing the effects of their mutations on RNA splicing/abundance.

See response to point 4.

(3) A comparison between loss-of-function phenotypes and loss-of-microexon splicing phenotypes could help interpret the findings from positive hits.

See response to points 3 and 4.

**Reviewer #2 (Public review):**
Summary:The manuscript from Calhoun et al. uses a well-established screening protocol to investigate the functions of microexons in zebrafish neurodevelopment. Microexons have gained prominence recently due to their enriched expression in neural tissues and misregulation in autism spectrum disease. However, screening of microexon functionality has thus far been limited in scope. The authors address this lack of knowledge by establishing zebrafish microexon CRISPR deletion lines for 45 microexons chosen in genes likely to play a role in CNS development. Using their high throughput protocol to test larval behaviour, brain activity, and brain structure, a modest group of 9 deletion lines was revealed to have neurodevelopmental functions, including 2 previously known to be functionally important.Strengths:(1) This work advances the state of knowledge in the microexon field and represents a starting point for future detailed investigations of the function of 7 microexons.(2) The phenotypic analysis using high-throughput approaches is sound and provides invaluable data.

We thank the reviewer for their support.

Weaknesses:(1) There is not enough information on the exact nature of the deletion for each microexon.

To clarify the nature of our mutant alleles, we have added a figure (Figure S1) that details the location of the microexon in relation to its predicted neighboring exons, deletion boundaries, guide RNAs, and putative regulatory elements.

(2) Only one deletion is phenotypically analysed, leaving space for the phenotype observed to be due to sequence modifications independent of the microexon itself.

We have determined whether the mRNA is impacted in unanticipated ways for a subset of mutants with mild phenotypes (see point #4 responses to Reviewer 1 for details). Our findings for three microexon mutants (ap1g1, vav2, and vti1a) are corroborated by LopezBlanch et al. (https://doi.org/10.1101/2024.10.23.619860). We have also already compared the microexon removal to a loss-of-function mutant for two lines (Figures S3 and S4), and we have made this comparison more obvious as well as increasing the discussion of the expected phenotypes from typical loss-of-function mutants (see point #3 response to reviewer 1).

Unlike protein-coding truncations, clean removal of the microexon and its regulatory elements is unlikely to yield different phenotypic outcomes if independent lines are generated (with the exception of genetic background effects). When generating a proteintruncating allele, the premature stop codon can have different locations and a varied impact on genetic compensation. In previous work (Capps et al., 2024), we have observed different amounts of nonsense-mediated decay-induced genetic compensation (El-Brolosy, et al., 2019) depending on the location of the mutation. As they lack variable premature stop codons (the expectation of a clean removal), two mutants for the same microexons should have equivalent impacts on the mRNA.

We now address the concern of subtle genetic background effects in the Methods: “Even with using sibling controls and collecting multiple biological replicates from individual parents, the possibility remains that linked genetic variation may have contributed to the mild phenotypes we observed, as only a single line was generated.”

**Reviewer #3 (Public review):**
Summary:This paper sought to understand how microexons influence early brain function. By selectively deleting a large number of conserved microexons and then phenotyping the mutants with behavior and brain activity assays, the authors find that most microexons have minimal effects on the global brain activity and broad behaviors of the larval fish-- although a few do have phenotypes.Strengths:The work takes full advantage of the scale that is afforded in zebrafish, generating a large mutant collection that is missing microexons and systematically phenotyping them with high throughput behaviour and brain activity assays. The work lays an important foundation for future studies that seek to uncover the likely subtle roles that single microexons will play in shaping development and behavior.

We thank the reviewer for their support.

Weaknesses:The work does not make it clear enough what deleting the microexon means, i.e. is it a clean removal of the microexon only, or are large pieces of the intron being removed as well-- and if so how much? Similarly, for the microexon deletions that do yield phenotypes, it will be important to demonstrate that the full-length transcript levels are unaffected by the deletion. For example, deleting the microexon might have unexpected effects on splicing or expression levels of the rest of the transcript that are the actual cause of some of these phenotypes.

To clarify the nature of our mutant alleles, we have added a figure (Figure S1) that details the location of the microexon in relation to its predicted neighboring exons, deletion boundaries, guide RNAs, and putative regulatory elements. We have determined whether the mRNA is impacted in unanticipated ways for a subset of mutants with mild phenotypes (see point #4 responses to Reviewer 1 for details).

**Reviewer #1 (Recommendations for the authors):**
(1) For most ME mutations, 4 guide sequences are provided. More description / a diagram could be helpful to interpret how ME mutations were generated.

We have added diagrams to the Supplementary Materials (new Figure S1) to show where the guide RNAs, cut sites, and putative regulatory elements are in relationship to the microexon and its neighboring exons. We have also added the following point to the text: “Four guide RNAs were used, two on each side of the microexon (Table S2, Figure S1).”

(2) Figure 1 indicates that there are 45 microexons (MEs) but the text initially indicates that there are 44 that exist in a canonical layout (the text later indicates there are 45). This could be made more clear.

The 45 refers to the mutants that were generated, not the microexons with putative Srrm4 regulatory elements. We did not choose microexons to mutate based on whether they were regulated by Srrm4. We have clarified these points in the manuscript as follows: “Of these 95 microexons, 42 exist in a canonical layout in the zebrafish genome, with both a UGC and UC repeat – or similar polypyrimidine tract – directly upstream of the alternatively spliced exon (Gonatopoulos-Pournatzis et al., 2018) (Table S1), indicating that Srrm4 likely controls their inclusion. Of the remaining microexons, 44 are organized similarly to the canonical layout, typically with either a UGC or UC repeat. Thus, they may also be regulated by Srrm4.” and “Using CRISPR/Cas9, we generated lines that removed 45 conserved microexons (Table S2) and assayed larval brain activity, brain structure, and behavior (Figure 1A). Four guide RNAs were used, two on each side of the microexon (Table S2, Figure S1). For microexons with upstream regulatory elements that are likely important for splicing, these elements were also removed (Figure S1).”

(3) The description of the "canonical layout" as containing TGC / UC repeats could be rewritten as either "containing a UGC motif and UC repeats" or "containing a TGC motif and TC repeats."

This error has been corrected.

(4) Why was tcf7l2 selected as a control for MAP mapping?

The mutant for tcf7l2 is an example of a moderately strong phenotype from a recent study we completed (Capps et al., 2025). This mutant was selected because it has both increased and decreased activity and structure and is ideal for setting the range of the graph. We now include a comparison to additional mutants from this study of autism genes (Capps et al., 2025) to further demonstrate how mild the phenotypes are in the microexon removal mutants (new Figure 3G). We also include the activity and structure maps of tcf7l2 mutants in Supplementary Figures 9 and 11.

(5) What does it mean that of the remaining microexons, most are similar to canonical layout?

Typically, they would have one of the two regulatory elements instead of both or the location of the possible elements would be slightly farther away than expected. We have clarified this point in the manuscript as follows: “Of these 95 microexons, 42 exist in a canonical layout in the zebrafish genome, with both a UGC and UC repeat or similar polypyrimidine tract – directly upstream of the alternatively spliced exon (Gonatopoulos-Pournatzis et al., 2018) (Table S1), indicating that Srrm4 likely controls their inclusion. Of the remaining microexons, 44 are organized similarly to the canonical layout, typically with either a UGC or UC repeat. Thus, they may also be regulated by Srrm4.”

(6) Figure 2A is very difficult to see - most are either up or down - suggest splitting into 2 figures - one = heat map, second can summarize values that were both up and down.

We prefer to retain this information for accuracy. The bubble location is offset to effectively share the box between the orange (decreased) and purple (increased) measures. For example, and as noted in the methods and now expanded upon, a measure can change between 4 and 6 dpf or a measure such as bout velocity could be increased while the distance traveled is decreased (both are magnitude measures). The offset of the bubbles is consistently 0.2 data units in x and y from the center of the box.

(7) The authors apply rigorous approaches to testing the importance of microexons. I especially appreciate the inclusion of separate biological replicates in the main figures!

We thank the reviewer for their positive feedback.

(8) Page 5 line 5 - suggest "compared to homozygous mutants".

The change has been made.

(9) For Eif5g3b dark flash phenotype, it's not clear what "p-values are not calculated for response plots" means. A p-value is provided in the plot for ppp6r3 response freq.

The eif4g3b plot is the actual response trace measuring through pixel changes whereas the ppp6r3 is the frequency of response. While informative, the response plot is time-based data with a wide dynamic range, making the average signal across the entire time window meaningless. We include the p-values for a related measure, the latency for the first 10 dark flashes in block 1 (day6dpfdf1a_responselatency) in the legend.

(10) The ptprd phenotype in 2D is not described in the text.

The change has been made.

(11) Page 7 line 7: "mild" is repeated.

This error has been corrected.

**Reviewer #2 (Recommendations for the authors):**
Specific points for needed improvement:(1) The title should be adjusted to more accurately describe the results. The term 'minimal' is under-representing the findings. 9/45 (20%) of targets in their screen have some phenotype, indicating that a significant number have indeed an important function. Moreover, the phenotypic analysis is limited, leaving space for missed abnormalities (as discussed by the authors). I would therefore suggest a more neutral title such as 'Systematic genetic deletion of microexons uncovers their roles in zebrafish brain development and larval behaviour'.

While some microexon mutants do have repeatable phenotypes, these phenotypes are far milder than phenotypes observed in other mutant sets. We now include a comparison to additional mutants from this study of autism genes (Capps et al.,2025) to further demonstrate how mild the phenotypes are in the microexon removal mutants (new Figure 3G). The title states that these microexons have a minimal impact on larval zebrafish brain morphology and function, leaving room for the possibility of adult phenotypes. Thus, we prefer to retain this title.

(2) Do the 45 chosen microexons correspond to the 44 with a canonical layout with TGC and UC repeats? If so, it needs to be explicitly stated in the text that exons were chosen for mutation based on the potential for SRRM4 regulation. If not, then the rationale for the choice of the 45 mutants from the 95 highly conserved events needs to be explained further.

The 45 refers to the mutants that were generated, not the microexons with putative Srrm4 regulatory elements. We did not choose microexons to mutate based on whether they were regulated by Srrm4. We have clarified these points in the manuscript as follows: “Of these 95 microexons, 42 exist in a canonical layout in the zebrafish genome, with both a UGC and UC repeat – or similar polypyrimidine tract – directly upstream of the alternatively spliced exon (Gonatopoulos-Pournatzis et al., 2018) (Table S1), indicating that Srrm4 likely controls their inclusion. Of the remaining microexons, 44 are organized similarly to the canonical layout, typically with either a UGC or UC repeat. Thus, they may also be regulated by Srrm4.” and “Using CRISPR/Cas9, we generated lines that removed 45 conserved microexons (Table S2) and assayed larval brain activity, brain structure, and behavior (Figure 1A). Four guide RNAs were used, two on each side of the microexon (Table S2, Figure S1). For microexons with upstream regulatory elements that are likely important for splicing, these elements were also removed (Figure S1).”

There was no clear rationale for those that were selected. We attempted to generate all 95 and some mutants were not successfully generated in our initial attempt. As we found minimal phenotypes, we elected to not continue to make the remaining ones on the list.

(3) More detail regarding the design of guides for CRISPR is required in the text in the methods section. From Table S2, 4 guides were used per microexon. Were these designed to flank the microexon? How far into the intronic sequence were the guides designed? Were the splicing regulatory sequences (polypyrimidine tract, branchpoint) also removed? The flanking sequences of each of the 45 deletion lines need to be provided.

We have added diagrams to the Supplementary Materials (new Figure S1) to show where the guide RNAs, cut sites, and putative regulatory elements are in relationship to the microexon and its neighboring exons. We removed the microexon and the surrounding area that contains the putative regulatory elements.

(4) Following on from the previous point, to ascertain that the phenotype observed is truly due to lack of microexon (rather than other event linked to removed intronic sequences) - for the 7 exons newly identified as functionally important, at least one added deletion line has to be shown, presenting the same phenotype. If making 7 more lines can't be achieved in a reasonable time (we are aware this is a big ask), a MO experiment blocking microexon splicing needs to be provided (may not be ideal for analysis at 6 dpf). For the existing mutants and the new ones (or morphants), sequencing of the mRNAs for the 7 genes in mutants and siblings also needs to be added to check any possible change in other variants.

Unlike protein-coding truncations, clean removal of the microexon and its regulatory elements is unlikely to yield different phenotypic outcomes if independent lines are generated (with the exception of genetic background effects). When generating a protein-truncating allele, the premature stop codon can have different locations and a varied impact on genetic compensation. In previous work (Capps et al., 2024), we have observed different amounts of nonsense-mediated decay-induced genetic compensation (El-Brolosy, et al., 2019) depending on the location of the mutation. As they lack variable premature stop codons (the expectation of a clean removal), two mutants for the same microexons should have equivalent impacts on the mRNA. We acknowledge that we inadequately described the generation of these alleles, and we now provide Figure S1 to show the microexon’s relationship to possible regulatory elements that impact splicing in unexpected ways if they remain.

We now acknowledge the concern of subtle genetic background effects in the Methods: “Even with using sibling controls and collecting multiple biological replicates from individual parents, the possibility remains that linked genetic variation may have contributed to the mild phenotypes we observed, as only a single line was generated.”

Given the caveats of MOs and transient microinjection for the study of 6 dpf phenotypes, we disagree that this suggested experiment would provide value. The phenotypic assays we use are highly sensitive, and we would not even trust CRISPANTs to yield reliable data. We have added an additional loss-of-function allele for ppp6r3 from the Sanger knockout project, which has a similar but stronger size change to the ppp6r3 microexon-removal line. In addition, our findings for three microexon mutants (ap1g1, vav2, and vti1a) are corroborated by Lopez-Blanch et al. (https://doi.org/10.1101/2024.10.23.619860).

To support that these we generated clean removal of these microexons, we experimentally determined whether the mRNA is impacted in unanticipated ways for a subset of mutants with mild phenotypes (see the point #4 public response to Reviewer 1). We also have already compared the microexon removal to a loss-offunction mutant for two lines (Figure S1), and we have made that outcome more obvious as well as increasing the discussion of the expected phenotypes from typical loss-of-function mutants (see point #3 public response to Reviewer 1).

(5) Figure 3: An image of control tcf7l2 mutant brain activity as a reference should be included.

We now include the activity and structure maps of tcf7l2 mutants in Supplementary Figures 9 and 11.

(6) Figure 3a/b. The gene names on the y-axis of the pERK and structure comparisons should be reordered to be alphabetical so that phenotypes can be compared by the reader for the same microexon across the two assays.

These data are clustered so that any similarities between maps can be recognized. We prefer to retain the clustering to compare lines to each other.

(7) Figure S6 legend. Including graph titles like "day3msdf_dpix_numberofbouts_60" is not comprehensible to the reader so should be replaced with more descriptive text. As should jargon such as "combo plot" and"habituation_day5dpfhab1post_responsefrequency_1_a1f1000d5p" etc.

The legend has been edited to describe the experiments. Subsections of the prior names are maintained in parentheses to enable the reader to connect the plots in this figure to the specific image and underlying data in Zenodo.

(8) Page 2 line 21 "to enable proper".

The change has been made.

(9) Page 7 line 7. Repeatable phenotypes were mild mild.

This error has been corrected.

**Reviewer #3 (Recommendations for the authors):**
(1) Figure 1B is confusingly laid out.

We are unclear how to modify Figure 1B, as it is a bar plot. We have modified several figures to improve clarity.

(2) Figure 1E-there are some pictures of zebrafish but to what end? They aren't labelled. The dark "no expression" looks really similar to the dark green, "high expression".

The zebrafish images represent the ages assessed for microexon inclusion. We have added labels to clarify this point.

(3) The main text says "microexons were removed by Crispr" but there is no detail in the main text about this at all-- and barely any in the methods. What does it mean to be removed? Cleanly? Or including part of the introns on either side? Etc. How selected, raised, etc? I can glean some of this from the Table S2 if I do a lot of extra work, but at least some notes about this would be important.

We have added diagrams to the Supplementary Materials (new Figure S1) to show where the guide RNAs, cut sites, and putative regulatory elements are in relationship to the microexon and its neighboring exons. We removed the microexon and the surrounding area that contains the putative regulatory elements.

(4) Figure 2 - There are no Ns, at least for the plots on the right. The reader shouldn't have to dig deep in Table S2 to find that. It is also unclear why heterozygous fish are not included in these analyses, since there are sibling data for all. Removed for readability of the plots might be warranted, but this should be made explicitly clear.

The Ns for these plots have been added to the legend. The legend was also modified as follows: “Comparisons to the heterozygous larvae are removed for clarity and available in the Supplementary Materials, as they often have even milder phenotypes than homozygous.”

(5) Needed data: for those with phenotypes, some evidence should be presented that the full-length transcripts that encode proteins without the microexons are still expressed at the same level and without splicing errors/NMD. Otherwise, some of these phenotypes that were found could be due to knockdown or LOF (or I suppose even overexpression) of the targeted gene.

We have added a new Supplementary Figure S2 confirming clean removal of the microexons with RT-PCR for a subset of mutants with phenotypes. This figure also includes qRT-PCR for the same subset. We now discuss these findings: Results: “For eight mutant lines, we confirmed that the microexon was eliminated from the transcripts as expected (Figure S2). Although our genomic deletion did not yield unexpected isoforms, qRT-PCR on these eight lines revealed significant downregulation for the homozygous vav2 mutant (Figure S2), indicating possibly complex genetic regulation.”